# Impact Evaluation of An Interdisciplinary Educational Intervention to Health Professionals for the Treatment of Mild to Moderate Child Malnutrition in Mexico: A Difference-in-Differences Analysis

**DOI:** 10.3390/healthcare10122411

**Published:** 2022-11-30

**Authors:** Sonia Sánchez-Encalada, Myrna Mar Talavera-Torres, Antonio R. Villa-Romero, Marcela Agudelo-Botero, Rosa María Wong-Chew

**Affiliations:** 1División de Investigación, Facultad de Medicina, Universidad Nacional Autónoma de Mexico, Mexico City 04510, Mexico; 2Hospital Pediátrico de Coyoacán, Mexico City 04000, Mexico; 3Centro de Investigación en Políticas, Población y Salud, Facultad de Medicina, Universidad Nacional Autónoma de Mexico, Mexico City 04510, Mexico

**Keywords:** malnutrition, health professionals, educational intervention, difference-in-differences analysis

## Abstract

The prevalence of undernutrition in Mexican children younger than 5 years old has been 14% since 2006. There are clinical practice guidelines for mild to moderate malnutrition in children in the Mexican health system; however, they are not applied. In addition, the knowledge and practices of health professionals (HP) to treat malnutrition in health centers are insufficient to perform adequate assessments and correct treatments. An impact evaluation of an interdisciplinary educational intervention was carried out on 78 HPs for the treatment of children with mild to moderate malnutrition of low resources, with 39 in the intervention group and 37 in the counterfactual group, estimated as the comparison group. A Food and Agriculture Organization (FAO)-validated questionnaire adapted to child malnutrition about knowledge, attitudes, and practices was applied before, after, and 2 months after a malnutrition workshop. The difference-in-differences analysis showed that the educational intervention group had a significant improvement in knowledge, attitudes, and practices before and after the intervention (grades of 54.6 to 79.2 respectively, *p* = 0.0001), compared with the comparison group (grades of 79.2 and 53.4, respectively, *p* = 0.0001), which was maintained over two months (grades of 71.8 versus 49.8, *p* = 0.0001, respectively). The multivariate analysis showed that the probability of improvement in learning by 30% was 95-fold higher in the educational intervention group versus the comparison group, OR = 95.1 (95% CI 14.9–603.0), and this factor was independent of sex, age, education, or hospital position. Despite the availability of clinical practice guidelines for the assessment and treatment for child malnutrition, education in malnutrition for HPs is effective and needed to achieve a significant improvement in children’s health.

## 1. Introduction

One of the global public health problems to solve urgently is child undernutrition (ChUN) [1]. In 2019, half of the worldwide deaths of children under five years old were due to malnutrition and associated diseases such as gastrointestinal and respiratory infections [2]. Focusing on nutritional well-being offers the opportunity to create synergy between public health and equity, according to the Sustainable Development Goals 2030 Agenda, especially health-related goals, such as zero hunger, health and wellness, and hygiene and sanitation, to promote nutritional resilience [1].

In Mexico, stunting (low height for age) in children under 5 years old is a disease with a prevalence of 14% that has not changed since 2006, according to the National Health and Nutrition Survey (ENSANUT) from 2006, 2012, 2016, and 2018 [3]. This chronic malnutrition affects the physical health, mental development, and emotional well-being, in some cases irreversibly, so its detection and treatment in early stages are necessary to prevent serious consequences [4]. The most prevalent forms of ChUN in Mexico City are mild to moderate where the clinical manifestations are few or imperceptible and often underestimated and not diagnosed [5]. Therefore, the proper knowledge of first contact health professionals (HP) in primary care is necessary to detect and treat these patients correctly and on time to prevent severe malnutrition [6].

The Food and Agriculture Organization (FAO) and the 2020 Global Health Report have proposed HP education as one of the better strategies to fight ChUN. Education has been considered an economic, viable, and sustainable strategy that empowers the participants. It can be extended as a network to generate changes in knowledge and reach several people at the same time [1,7,8]. In the year 2004, Anand et al. observed that knowledge and practices of HP were insufficient to assess the correct treatment of child malnutrition (TCM) in health centers. In addition, in the year 2015 [9], Restier et al. (2015) showed that HP underestimate the prevalence of malnutrition in children by half and overestimate the frequency of appropriate screening practices and its detection [10]. On the other hand, in 2019, Nimpagaritse et al. agreed that TCM in health centers is deficient and that clinical practice guidelines are not applied effectively, which causes lower detection and deficient attention [11].

Evidence has shown that HP education is an indispensable strategy to improve their knowledge, attitudes, and practices; its effect has positive impacts on improving care and the health of the patients. In the year 2021, a study carried out in South Africa evaluated a mentoring intervention for HP for the treatment of HIV and the improvement of infant feeding. This group observed that participation in a multidisciplinary training team was associated with an attitude score improvement of five points, significantly higher in the intervention group compared to the control group [12]. However, in Mexico, there is no evidence of impact evaluations of educational programs for HP to treat mild to moderate undernutrition. Currently, it has been proposed that action-oriented nutrition education is effective to promote significant learning [13].

The aim of this study is to evaluate the impact of an interdisciplinary educational intervention on HP for the treatment of mild to moderate ChUN in Mexico City. The obtained evidence will be useful for decision makers to analyze the possibility of implementing educational programs on primary healthcare HP about ChUN. This could improve their knowledge, attitudes, and practices and therefore help them give proper care on time and give assertive recommendations to the mothers.

## 2. Material and Methods

### 2.1. Study Setting and Population

An impact evaluation was performed at the Hospital Pediátrico de Coyoacán during the period of January to June of 2022. This institution is a pediatric hospital that gives attention to children from very low-income families from the south of Mexico City. It was chosen because it serves as a primary and secondary care and treats children without severe complications [14]. The study population invited to participate included HP of any age with different positions at the institution, such as attending physicians, medical residents, interns, students, nurses, and nutritionists who were involved in the health care of the children (approximately 141 HP). Administrative, technical, and quartermaster personnel were excluded

### 2.2. Procedures

An impact evaluation to assess a social health program in Mexico and to provide information to decision-makers allowing improvements in implementation was carried out in two stages:

Stage I. Management.

First, the study was submitted for approval by the hospital authorities and by the teaching department. This research group designed an educational intervention based on the taxonomy of meaningful learning, and a theory of change (ToC) was developed to achieve the desired result, with intermediate and long-term objectives, assumptions, and risks. Subsequently, a FAO knowledge, attitudes, and practices (KAP) survey for health care providers and other personnel regarding HIV patients [15] was semantically and constructively adapted to the treatment of child malnutrition (Section 2.4) and reviewed by three experts; pertinent changes were made according to their observations.

Stage II. Implementation.

The program was scheduled with the teaching department, and all attending physicians, residents, interns, students, nutritionists, and nurses were invited to participate in the study. The assignment was not randomized; those who were available to take the workshop constituted the intervention group (IG), and the counterfactual group was estimated as the comparison group (CG), which included those who only answered the questionnaire. The KAP questionnaire was applied to both groups as the baseline or diagnostic measure, and informed consent was signed. A two-hour active interdisciplinary medical and nutritional workshop was given to the IG by a DSc. pediatrician and an MSc. nutrition education specialist. At the end of the workshop, the KAP questionnaire was applied again to both groups, and to observe the effect over time, the KAP questionnaire was applied to both groups two months after the intervention. The malnutrition workshop was given twice to reach the sample size calculated for the study. 

### 2.3. Theory of Change (ToC)

A ToC is an impact evaluation essential step that outlines how the project is supposed to achieve the intended results. It is a flexible, non-linear map that describes a results chain as a useful tool, specifies the evaluation question(s), and selects indicators to assess performance. At the beginning of the educational intervention design, a ToC map to describe and evaluate the approach of the intended outcomes was constructed. The impact the study wanted to achieve, located outside the ceiling of accountability (level that cannot be measured and therefore is outside the responsibility of the intervention outcome), was to improve HP undernutrition care to decrease mild to moderate undernutrition in Mexican children younger than 5 years old. The long-term outcome was to improve HP undernutrition care in the IG compared to the CG. The intermediate outcomes were to produce changes in the KAPs of HPs to treat undernutrition and give correct counseling to their mothers through a significant learning strategy by attending a 2-h active workshop. The principal indicator was the information collected in the KAP questionnaire before, after, and two months after the educational intervention in both groups. The rationale was that HPs needed to be updated about mild to moderate child undernutrition screening, medical treatment, nutritional care, and counseling to the mothers in order to be detect and treat undernutrition properly. Finally, we assumed that all HPs would answer the questionnaire before, after, and two months after the intervention and remain at the institution [16] (Figure 1).

### 2.4. Knowledge, Attitudes, and Practices Questionnaire Adapted for Malnutrition for Health Professionals (KAP)

A survey instrument for mild to moderate undernutrition was developed based on the validated FAO instruments of adaptable KAP model questionnaires using appendix 6, module 5. These questionnaires were already validated and needed to be adapted to the local context and the requirements of the specific project or intervention. The adaptation was constructed conceptually based on the KAP-FAO/WHO guidelines on malnutrition and food [15], on the KAP survey for healthcare providers and other professionals in relation to HIV/AIDS [17], and on the WHO/UNICEF reports, specific to the treatment of malnourished children [18]. To evaluate the comprehensibility of the instrument, the draft was evaluated by three experts in malnutrition (outside the study population), and open comments were obtained. The questionnaire was anonymous and was made up of 55 closed questions (true/false, Yes/No, don’t know answers); it also included a clinical case to evaluate the classification and diagnosis of the nutrition status according to the Z-score and recommendations to mothers. It did not include psychometric characteristics. An individual identification number was created for each participant, and general information such as sex, age, educational level, and position at the institution were also included; 39 questions evaluated knowledge, 6 evaluated practices, and 6 evaluated attitudes (51 total items). To grade the questionnaires, 1 was designated for the correct answers and positive KAP, and 0 was designated for incorrect or negative ones. The rating was calculated using a scale from 0 to 100, where 100 was the maximum grade obtained of the total of correct answers (51 items).

### 2.5. Educational Intervention

We designed an action-oriented, face-to-face workshop. It consisted of a one-day, two-hour session of two disciplines: medical and nutritional. The main objectives of this intervention were that HPs acquire knowledge, positives practices, and attitudes for the adequate treatment of children with mild to moderate malnutrition and give assertive recommendations to their mothers based on Dr. Fink’s meaningful learning taxonomy. Meaningful learning refers to the learning that causes a lasting change in the student and that becomes important for their life, promoting their development of critical and practical thinking through the dynamics and the resolution of a clinical case [13]. The topics included in the workshop were: the general context of malnutrition, recognition of the malnourished child, risk factors, classification according to the Z-score and diagnosis, correct diet for children with mild to moderate malnutrition, and recommendations to mothers according to the clinical practice guidelines for the diagnosis and treatment of malnutrition in children under five years of age at primary care; they were prepared with the participation of the National Health System Institutions and coordinated by the Technological Excellence National Center (GPC—SS-119-18) [19]. A team dynamic was implemented using specific educational material for food orientation based on the Equivalents Mexican System.

### 2.6. Sample Size Calculation

The sample size was calculated using the Epidat app, version 4.2, for the difference of proportions formula, considering Zα = 1.96, and Zβ = 0.84. The main outcome variable was the difference of 30% before and after the intervention [20]. A total of 78 participants were included, 39 in each group (intervention and comparison group). 

### 2.7. Statistical Analysis

The data was analyzed with the SPSS program, version 25. To show possible differences in knowledge, attitudes, and practices of the 78 health professionals, a difference-in-differences analysis was performed using specific statistical tests depending on the variable distribution. We used the Student’s *t*-test or Wilcoxon’s *t*-test for related samples and the Student’s *t*-test or a Mann–Whitney U test between groups. Finally, for the evaluation over time (two months after the intervention), an ANOVA of repeated measures or Friedman’s test was used. To estimate the probability of improvement in learning within the IG and the possible association with the co-variables (position in the institution, sex, level of education, age), multivariate logistic regression models were carried out, where the OR (odds ratio) was obtained, derived from the exponential of regression coefficients. 

#### Difference-in-Differences Estimation

To determine the effect of the educational intervention on the KAP variables, we used the difference-in-differences (DinD) analysis using specific statistic tests. DinD models compare changes in outcomes that occur due to an intervention compared to changes that would normally occur over time in the absence of an intervention using a comparison group. The following equation was used to calculate the DinD impact estimate: (DD) = (B − A) − (D − C), where (B − A) is the difference in outcomes for the IG, and (D − C) is the difference in outcomes for the CG [21]. 

### 2.8. Ethical Considerations and Informed Consent

The study was approved by the Research and Ethics Committees of the Faculty of Medicine of the Universidad Nacional Autónoma de México (#059/2014) and by the Ethics and Research Committees of the Coyoacán Pediatric Hospital (#3020010514). Confidentiality of personal data was maintained throughout the study by making participants’ information anonymous and asking them to provide honest answers. The participation of the HPs was voluntary and uncompensated. The informed consent was displayed on the initial page of the survey, read, and signed. The study was carried out according to the Declaration of Helsinki. As significant differences were found in the intervention group, the same workshop was given to the comparison group once the study was over.

## 3. Results

Seventy-eight HPs were included in the study. Out of 78 participants, 76 correctly responded to the questionnaire, with a response rate of 97.4%. To estimate the impact of the educational intervention, we compared the outcomes of the IG with 39 HPs with the estimation of the counterfactual group obtained from 37 HPs as the CG.

### 3.1. Demographic Characteristics

The baseline characteristics in both groups were not statistically different, so equal groups were assumed. The mean age was 32.2 ± 13.8 for the IG and 34.7 ± 13.2 for the CG. Of the participants, 7 (17.9%) were men and 32 (82.1%) were women in the IG, and 7 (18.9%) were men and 30 (81.1%) were women in the CG (*p* = 0.57). Regarding the educational level, the majority were undergraduate (30 (76.9%) for the IG and 29 (78.4%) for the CG), and 15 were postgraduate, 7 (17.9%) for the IG and 8 (21.6%) for the CG (*p* = 0.36). In regards to the hospital positions, the majority were nurses, 11 (28.2%) for the IG and 14 (37.8%) for the CG, and attending physicians, 4 (10.3%) for the IG and 8 (21.6%) for the CG (*p* = 0.34) (Table 1).

### 3.2. Knowledge, Attitudes, and Practices of Health Professionals on the Treatment of Moderate to Mild Undernutrition in Children and Recommendations to Their Mothers

We calculated the differences in the grades of the complete questionnaires (KAP) by sections before and after the intervention and between the intervention and the comparative groups using the Student’s *t*-test, Wilcoxon, or Mann–Whitney U tests, depending on the sample distribution. We found significant differences in all the characteristics before and after the workshop on KAP, with grades of 54.6 ± 6.6 before to 79.2 ± 8.2 after the workshop in the IG vs. 51.6 ± 8.2 to 53.4 ± 11.4 in the CG (*p* = 0.0001). In the analysis of the KAP questionnaire by sections, the grades for knowledge were 50.3 ± 8.8 before the workshop to 77.9 ± 10.3 after the workshop in the IG (*p* = 0.0001) vs. 47.7 ± 9.5 to 50.6 ± 12.7 in the CG; for practices, the grades were 38.4 (0–100) before to 88.0 (33–100) after the workshop in the IG (*p* = 0.0001) vs. 34.2 (0–100) to 35.1 (0–100) in the CG; finally, for attitudes, the grades were 85.0 (50–100) before to 89.3 (66–100) after the workshop in the IG (*p* = 0.0001) vs. 77.4 (33.1–100) to 72.9 (16.6–100) in the CG.

To calculate the DinD outcomes two months later, we used the ANOVA or the Friedman tests depending on the variable distribution. Statistically significant differences were found in all the characteristics in the baseline and in the 2 month grades before and after the workshop: 50.3 ± 8.8 to 71.7 ± 5.6 in the IG (*p* = 0.0001) vs. 47.7 ± 9.5 to 43.3 ± 4.8 in the CG for knowledge; 38.4 (0–100) to 68.8 (33–100) in the IG (*p* = 0.0001) vs. 34.2 (0–100) to 33.3 (33–33) in the CG for practices; and 85.0 (50–100) to 92.2 (83–100) in the IG (*p* = 0.0001) vs. 77.4 (33.1–100) to 78.5 (16.6–3.3) in the CG for attitudes. In the comparison between groups, the paragraph describes the grades before, after and 2 months after the intervention, and it also says that there were differences between the IG and CG: 71.7 ± 5.6 in the IG vs. 44.3 ± 4.8 in the CG (*p* = 0.0001) for knowledge; 68.8 (33–100) in the IG vs. 33.3 (33–33) in the CG (*p* = 0.0001) for practices; and 92.2 (83–100) in the IG vs. 78.5 (16.6–3.3) in the CG (*p* = 0.0001) for attitudes (Table 2). Table 3 summarizes the DinD analysis in the overall KAP questionnaire according to the previously described formula (Table 3).

### 3.3. Multivariate Analysis

Given the significant associations in the bivariate analysis, a multivariate logistic regression analysis was performed. The multivariate model is shown in Table 4, with a 30% cut-off point of learning. In terms of personal variables, it was found that the probability of improving in learning by 30% was 95-fold higher in the intervention group with regards to the comparison group, OR = 95.07 (95% CI 14.9–603.0), and this factor was independent of sex, age, education level, or hospital position (Table 4).

## 4. Discussion

Undernutrition in children under 5 years of age is a constant public health problem worldwide. In several countries, the double and triple burden of malnutrition is observed even within the same family; the trend is clear, despite the actions that are being carried out, and progress is slow to achieve well-being for all, according to the 2030 agenda [1].

Since the 2000s, the Pan American Health Organization (PAHO) has highlighted the need to educate HPs because the knowledge acquired at school tends to become obsolete. Therefore, a theoretical–practical education became essential for the improvement of competencies through a permanent health education program coordinated by the health centers’ human resources departments [22]. On the other hand, PAHO/WHO recognized that primary healthcare (PHC) constituted a central element in the health policies of Latin American countries as the main strategy to achieve some of the Millennium Development Goals (zero hunger, reduce poverty, eliminate acute and chronic malnutrition, and reduce infant and maternal mortality) [6]. They also emphasized that HP education could be an effective strategy to improve PHC services in malnourished children [23].

Nutrition is an important component of healthcare; several studies have concluded that offering nutritional treatment as part of the medical query is essential, and that it can have a positive impact on the health of the patients [24]. In 2016, Mogre et al. showed the importance of several educational interventions in healthcare systems that improved patient health by helping health students and professionals appreciate the importance of delivering nutrition care and to feel competent enough to deliver it [25]. This relates to Bzowyckyj et al. and Sharma et al., who showed that HP education programs can have direct impacts on patients and communities, as well as on HP learners themselves [26,27]. In Mexico City (2017), a quasi-experimental study demonstrated that an educational intervention by health professionals to low-income mothers with undernourished children improved the Z-score for weight/age after 6 months after the intervention with the same family resources [28]. However, in Mexico, there are no educational programs for HPs to treat child undernutrition. The current Mexican social programs are focused on attending undernutrition through monthly cash transfers or basic food baskets to families, monthly medical consultations, or sporadic food orientation talks in healthcare centers [29]. Hence, it is important to plan a good strategy that increases the capacities of the HPs to be able to attend the needs of these families.

The care expected from PHC workers includes nutrition assessment, education, and counseling interventions, monitoring, and evaluation [30]. Given those requirements, we adapted a FAO-KAP questionnaire for HPs who worked in a pediatric hospital that provided PHC in the south of Mexico City. Then, we designed a medical and nutritional workshop about the diagnosis and treatment of childhood mild to moderate undernutrition and about counseling low-income parents about ongoing care for their children. We conducted an impact evaluation to assess an educational intervention through scores obtained on the KAPs of the health professionals who participated in the workshop.

At the baseline measure, we confirmed that the KAPs of the HPs were insufficient to detect and treat child undernutrition, as well as to give proper counsel to parents, probably because health professionals are not familiar with, nor do they apply the specific guidelines, to treat children suffering from undernutrition. This relates to Gera et al., who observed that maternal care-seeking behavior could be more appropriate with the implementation of an integrated management of a childhood illness strategy (IMCI) [31]. We observed that the KAPs of HPs improved by 25.8 points after the educational intervention (grades of 53.4 to 79.2) compared to the CG (grades of 54.5 to 51.6)); the difference was statistically significant in the IG. This effect persisted after two months (grade of 71.8). Moreover, the DinD analysis showed that between groups, there was a statistically significant 22.8-point difference, suggesting that this impact can be attributed to the educational intervention. These results showed that KAPs about child undernutrition in HPs can be improved with an interactive multidisciplinary workshop based on meaningful learning and persist over time. This is in agreement with Sunguya et al. (Japan 2013), who showed in a systematic review of nutrition education provided to doctors and nurses on the diagnosis and treatment of malnourished children that training HPs improved knowledge, recommendations, and clinical practice, and it was reflected in the health improvement of malnourished children [32]. It is also in agreement with Mogre et al., who showed in a systematic review of 46 papers that educational interventions of HPs improved skills, attitudes, knowledge, and clinical practices [33].

In addition, we observed that the educational intervention improved KAPs in all the health staff, according to that observed by Sumantra-Ray et al. (England, 2014) and Levy et al. (2010). They agreed that a nutrition intervention to all educational levels was useful to improve the nutritional knowledge and skills of junior doctors, assistant physicians, nurses, and others and reported enthusiasm for the hospital setting, increasing awareness of clinical and public health nutrition outcomes [34,35]. Despite the statistically significant difference in the results, we observed that over time, the trend of knowledge began to decrease, possibly due to basic concept oblivion, so we agree with Huang et al. [36] that nutrition education programs offer a practical approach to continuing education for HPs; they may help eliminate the barriers to providing nutrition in primary care, such as lack of knowledge, skills, and self-confidence to give counseling, as well as eliminate negative attitudes towards the delivery of nutrition care [37,38,39].

The significant OR of 95 found in the KAPs of HPs in the multivariate analysis, which was independent of sex, age, educational level, and hospital position, could be due to the study design, to the interdisciplinary intervention, and to the teaching strategy delivered by expert HPs. Our results emphasized those of Robert et al., who showed that strengthening the interaction between HPs and caregivers is necessary to prevent malnutrition, responding to the context and culture of the community [40]. In addition, we agree with Crowley et al., who showed in a systematic review that nutrition is insufficiently incorporated into medical education, regardless of country, setting, or year of medical education [41].

One of the benefits of impact evaluations is that they provide credible evidence on the program performance and effectiveness; our results suggested that the design and delivery achieved the desired outcome: improvement of the KAPs of HPs; that can be attributed to the educational intervention. Nevertheless, our study had limitations; we cannot assume that our intervention would be effective in other institutions. The sample size and only one hospital setting reduces the possibility of contrasting with other types of health institutions and HP competences, which may be affected by other environmental variables.

## 5. Conclusions

We conclude that the current KAPs about undernutrition in children in primary healthcare by HPs are not adequate. An educational workshop provided to HPs about child undernutrition is an essential strategy to improve KAPs that could result in better nutrition care and better counseling to the mother. The particular characteristics of the intervention, such as geographic region, focus on meaningful learning, and the action-oriented, interdisciplinary nature of the workshop were relevant to the improvement of the KAPs of the HPs. In addition, we conclude that the intervention is equitable and inclusive to all heath staff, and we suggest it should be part of a continuing education program in primary healthcare centers around the country.

## Figures and Tables

**Figure 1 healthcare-10-02411-f001:**
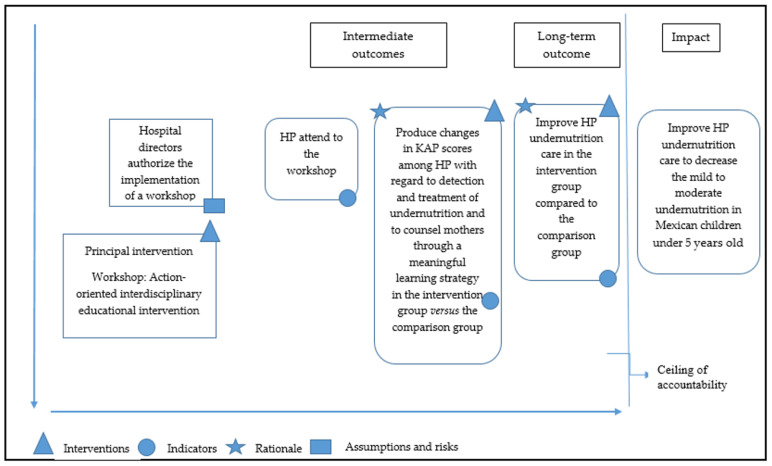
Theory of change. Indicator: The information collected in the KAP questionnaire before, after and two months after the educational intervention in both groups. Rationale: HP needed to be educated about mild to moderate children undernutrition screening, treatment, nutrition care and counseling to the mothers. Assumptions and risks: HP would answer the questionnaire before, after and two months after the intervention and remain in the institution. Source: Own elaboration based on Mary De Silva’s theory of change practical guide (2014).

**Table 1 healthcare-10-02411-t001:** Demographic characteristics of the population by group.

Characteristics	Intervention Group	Comparative Group	*p*
	*n* = 39	*n* = 37	
**Age (years), mean ± SD**	32.2 ± 13.8	34.7 ± 13.2	0.42
**Variable, *n* (%)**			
**Sex**			0.57
Men	7 (17.9)	7 (18.9)	
Women	32 (82.1)	30 (81.1)	
**Educational level**			0.36
Bachelors degree	2 (5.1)	0	
Undergraduate	30 (76.9)	29 (78.4)	
Postgraduate	7 (17.9)	8 (21.6)	
**Hospital position**			0.34
Nursing assistant	9 (23.1)	5 (13.5)	
Nurse staff	11 (28.2)	14 (37.8)	
Interns	10 (25.6)	5 (13.5)	
Residents	5 (12.8)	5 (13.5)	
Attending physicians	4 (10.3)	8 (21.6)	

**Table 2 healthcare-10-02411-t002:** Difference-in-differences estimation in knowledge, attitudes, and practices (KAP).

Characteristic Grades	Intervention Group *n* = 39	*p*	Comparative Group *n* = 37	*p*	Difference between Groups
	mean ± SD		mean ± SD		
**KAP**					
Baseline	54.6 ± 6.6		51.6 ± 8.2		
After the intervention	79.2 ± 8.2	0.0001 *,+	53.4 ± 11.4	0.24	25.8
After two months **	71.8 ± 11	0.0001 *,&	49.8 ± 3.5		
**Knowledge**					
Baseline	50.3 ± 8.8		47.7 ± 9.5		
After the intervention	77.9 ± 10.3	0.0001 *,+	50.6 ± 12.7	0.21	27.3
After two months **	71.7 ± 5.6	0.0001 *,&	44.3 ± 4.8		
	Median (min–max)		Median (min–max)		
**Nutritional knowledge**					
Baseline	66.6 (33–100)		66.6 (16.6–83)		
After the intervention	83.3 (50–100)	0.0001 *,++	66.6 (16–100)		16.7
After two months **	83.3 (33–100)	0.0001 *,/	65.0 (16.6–66)	0.30	
**Diagnosis, classification of undernourished children, and clinical case**					
Baseline	13.6 (0–66.6)		17.1 (0–100)		
After the intervention	83.7 (0–100)	0.0001 *,++	21.6 (0–100)		62.1
After two months **	92.8 (33–100)	0.0001 *,/	55.5 (33–100)	0.19	
**Recommendations to mothers**					
Baseline	19.4 (0–60)		17.3 (0–100)		
After the intervention	78.9 (0–100)	0.0001 *,++	24.8 (0–100)		54.1
After two months **	74.1 (40–100)	0.0001 *,/	28.0 (20–60)	0.82	
**Practices**					
Baseline	38.4 (0–100)		34.2 (0–100)		
After the intervention	88.0 (33–100)	0.0001 *,++	35.1 (0–100)		52.9
After two months **	68.8 (33–100)	0.0001 *,/	33.3 (33–33)	1.00	
**Attitudes**					
Baseline	85.0 (50–100)		77.4 (33.1–100)		
After the intervention	89.3 (66–100)	0.0001 *,++	72.9 (16.6–100)		16.4
After two months **	92.2 (83–100)	0.0001 *,/	78.5 (16.6–3.3)	1.00	

** Information of 7 participants of the comparative group and 15 of the treatment group. * *p* < 0.05. +: t-student; ++: Wilcoxon; &: ANOVA; /: Friedman tests.

**Table 3 healthcare-10-02411-t003:** DinD calculation in the complete KAP questionnaire.

Characteristic	After	Before	Difference
Intervention group	79.2	53.4	79.2 − 53.4 = 25.8 *
Comparison group	54.6	51.6	54.6 − 51.6 =3.0
Difference	24.6 *	1.8	25.8 − 3.0 = 22.8 *

* *p* < 0.05.

**Table 4 healthcare-10-02411-t004:** Logistic multivariate analysis adjusted by sex, age, educational level, and hospital position to estimate the probability of improving in learning.

Variable	OR	(95% CI)	*p*
**Group** Comparison Intervention	1.0 95.0	- 14.9–603.0	- 0.0001 *
**Sex** Male Female	1.0 1.5	- 0.3–9.6	- 0.65
Age (years)	1.0	1.0–1.1	0.67
**Educational level**			
Undergraduate	0.2	0.0–6.0	0.18
Postgraduate	0.3	0.0–9.6	0.51
**Hospital position**			
Nurses	3.8	0.0–28.0	0.54
Interns	0.5	0.0–24.5	0.72
Residents	0.8	0.0–61.2	0.91
Attending physicians	1.3	0.0–52.9	0.89

* *p* < 0.05.

## Data Availability

The data presented in this study are available on request from the corresponding author. The data are not publicly available due to privacy issues.

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
