# Peer review of "Impact Evaluation of An Interdisciplinary Educational Intervention to Health Professionals for the Treatment of Mild to Moderate Child Malnutrition in Mexico: A Difference-in-Differences Analysis"

_healthcare, 2022, doi:10.3390/healthcare10122411_

Round 1

Reviewer 1 Report

A brief summary

The manuscript is interesting and suitable for the selected journal. The text though is more difficult to read because of the English language. It needs to be checked by a professional before publication.

General concept comments

The introduction adequately describes the issue, stating the most important facts and emphasizing the aim of the study. The chapter on Materials and methods is extensive and describes the work flow in detail.

It is suggested not to repeat and describe the results that have already been presented, possibly those that are more important for the article.

Do not use a semicolon unless it’s absolutely necessary.

Please include below every Table what * means.

Always use the same number of decimal places for numbers in text and tables.

When citing the references, please address to Instruction for authors.

The conclusion is incomplete and too general. It needs to be expanded on why authors came to this conclusion (please, briefly state what You got from the research).

Specific comments

Line 33-34            Please correct the keywords and put “malnutrition” instead “mild to moderate malnutrition”.

Line 48                  Instead “so it is necessary its detection” please put “so its detection and treatment in early stages are necessary…”

Line 49-57            Please rewrite those sentences to make them more understandable. If it is necessary, include shorter sentences.

Line 62                  Perhaps “the treatment of child undernutrition” is a better term

Line 65-67            Please rewrite the sentence to make it more understandable.

Line 68                  Instead just “2021” put “in the year 2021”

Line 77                  “Obtained evidence will be useful” instead “The evidence generated”

Line 77-80            Too long sentence. Please make it clearer.

Line 273-286       The entire 2 paragraphs in the discussion belong to the Introduction. In the Discussion section, obtained results should be discussed and compared to other research. Please address to Instruction for authors.

Line 287-289       Please rewrite the sentence to make it clearer or just make several sentences.

Line 374                More than half of the references are older than 5 years. Authors are recommended to revise the last section References and add newer literature, no older than 5 years or replace existing literature with newer ones.

Line 414-415       Please include a year of the reference

Reviewer 2 Report

In the introduction and in the results, child undernutrition is repeated several times, it would be convenient to put its acronym the first time and use it in the following ones.

The AIMS paragraph should be reorganized, indicating first the need for the proposed interventions and then the purpose of the paper.

It is not explained what the adaptation of the KAP questionnaire consisted of, nor if the validation was semantic or constructive, nor its psychometric characteristics.

In line 109, name the intervention group as IG and the control group as CG, and use these acronyms in lines 120-130, 209-210

Figure 1 can be understood as a flow chart, and it is not well represented at what time the educational intervention was carried out.

Sample size: the population that could participate in the study is not specified

A Statistical Analysis section should be included to describe said analysis and not to do so in the presentation of results.

The description of tables 2 and 3 is confusing, it should be done individually

In tables 3 and 4 the meaning of * is not indicated.

The findings in Table 3 are not discussed.

BIBLIOGRAPHY:

Writ the acronym the first time and use it in the following ones.

It is recommended that when citing the bibliography, it be done according to Vancouver

In the reference 21 possibly missing data

30% of the bibliography is prior to 2012
